# Events Leading to the Establishment of Pregnancy and Placental Formation: The Need to Fine-Tune the Nomenclature on Pregnancy and Gestation

**DOI:** 10.3390/ijms242015420

**Published:** 2023-10-21

**Authors:** Giuseppe Benagiano, Salvatore Mancuso, Sun-Wei Guo, Gian Carlo Di Renzo

**Affiliations:** 1Faculty of Medicine and Surgery, Sapienza University of Rome, 00185 Rome, Italy; pinoingeneva@bluewin.ch; 2Geneva Foundation for Medical Education and Research, 1206 Geneva, Switzerland; 3Faculty of Medicine and Surgery, Catholic University of the Sacred Heart, 00168 Rome, Italy; salmancuso71@gmail.com; 4Research Institute, Shanghai Obstetrics & Gynecology Hospital, Fudan University, Shanghai 200011, China; hoxa10@outlook.com; 5Center for Perinatal and Reproductive Medicine, University of Perugia, 06156 Perugia, Italy; 6Department of Obstetrics, Gynecology and Perinatology, I.M. Sechenov First Moscow State Medical University, 119146 Moscow, Russia; 7Department of Obstetrics & Gynecology, Wayne State University School of Medicine, Detroit, MI 48201, USA

**Keywords:** early embryonic loss, embryo-derived platelet-activating factor, early pregnancy factor, gestation, pregnancy, pre-implantation factor

## Abstract

Today, there is strong and diversified evidence that in humans at least 50% of early embryos do not proceed beyond the pre-implantation period. This evidence comes from clinical investigations, demography, epidemiology, embryology, immunology, and molecular biology. The purpose of this article is to highlight the steps leading to the establishment of pregnancy and placenta formation. These early events document the existence of a clear distinction between embryonic losses during the first two weeks after conception and those occurring during the subsequent months. This review attempts to highlight the nature of the maternal–embryonic dialogue and the major mechanisms active during the pre-implantation period aimed at “selecting” embryos with the ability to proceed to the formation of the placenta and therefore to the completion of pregnancy. This intense molecular cross-talk between the early embryo and the endometrium starts even before the blastocyst reaches the uterine cavity, substantially initiating and conditioning the process of implantation and the formation of the placenta. Today, several factors involved in this dialogue have been identified, although the best-known and overall, the most important, still remains *Chorionic Gonadotrophin*, indispensable during the first 8 to 10 weeks after fertilization. In addition, there are other substances acting during the first days following fertilization, the *Early Pregnancy Factor*, believed to be involved in the suppression of the maternal response, thereby allowing the continued viability of the early embryo. The *Pre-Implantation Factor,* secreted between 2 and 4 days after fertilization. This linear peptide molecule exhibits a self-protective and antitoxic action, is present in maternal blood as early as 7 days after conception, and is absent in the presence of non-viable embryos. The *Embryo-Derived Platelet-activating Factor*, produced and released by embryos of all mammalian species studied seems to have a role in the ligand-mediated trophic support of the early embryo. The implantation process is also guided by signals from cells in the decidualized endometrium. Various types of cells are involved, among them epithelial, stromal, and trophoblastic, producing a number of cellular molecules, such as cytokines, chemokines, growth factors, and adhesion molecules. Immune cells are also involved, mainly uterine natural killer cells, macrophages, and T cells. In conclusion, events taking place during the first two weeks after fertilization determine whether pregnancy can proceed and therefore whether placenta’s formation can proceed. These events represent the scientific basis for a clear distinction between the first two weeks following fertilization and the rest of gestation. For this reason, we propose that a new nomenclature be adopted specifically separating the two periods. In other words, the period from fertilization and birth should be named “gestation”, whereas that from the completion of the process of implantation leading to the formation of the placenta, and birth should be named “pregnancy”.

## 1. Introduction

A just-published “Perspective” article in the American Journal of Obstetrics and Gynecology [1] proposes to make a distinction between “*Gestation*” (the period going from fertilization of the oocyte and birth), and “*Pregnancy*” (the period between the completion of the implantation process and birth). Biologically, such a distinction is justified by the fact that, in humans, there is strong and diversified evidence that at least 50% of early embryos do not proceed beyond the pre-implantation period. To consider these as “pregnancies” would imply that an attempt should be made to salvage these early embryos [2]. Such an attempt would be not only impossible, but also totally counterproductive, since today we know that the loss of preimplantation embryos is biologically a welcome and desirable process, given the high proportion of abnormalities in them and the need for a quality-control checkpoint before proceeding to a highly energy-consuming and somewhat precarious process.

The subject has been amply debated and reported, but—given the proposed new terminology—revisiting and summarizing what we know about the events occurring during the first two post-fertilization weeks, seems useful and timely.

In 2002 a review entitled “*Conception to Ongoing Pregnancy: The ‘Black Box’ of Early Pregnancy Loss*” [3] pointed out that preclinical loss of embryo, rather than failure of conception, accounted for much of human low fertility. Several years later, another review tried to evaluate the proportion of zygotes and early human embryos that under physiological conditions proceed to term [4]. Since then, not only new information coming from a number of different sources has been gathered, but also vital evidence has been produced on mechanisms leading to early embryonic losses.

A well-documented review of different new aspects of the processes leading to the loss of early embryos has been recently published, focusing on the growing understanding of the dialog established between the embryo and the endometrium [5]. It provided new evidence that an active selection aimed at impeding implantation of unhealthy embryos actually occurs at the endometrial interface, replacing the classic concept of mere ‘receptivity’ with one implying ‘selectivity’ by the host organism.

Worth mentioning is the finding that pregnancy wastage may be a function of the time-lapse between ovulation and implantation, with a decreasing probability of successful nidation with increasing peri-implantation time [4]. In addition, it has been shown that cycles in which intercourse occurred during the implantation window were significantly less likely to result in a positive pregnancy test after adjusting for major variables [Fecundability Ratio = 0.62, 95% Confidence Interval (CI) = 0.42–0.91] [6].

These views ran contrary to the prevailing common wisdom which, until the middle of the 20th century, dictated that the vast majority of human zygotes are normal and proceed to term, since congenital anomalies at birth are fortunately rare, with an estimated incidence of 6% (with a caveat that in a number of cases, statistics are not adjusted for pregnancy terminations) [7]. Indeed, once pregnancy has been confirmed, the overall rate of spontaneous abortion does not exceed 15%, as documented by a large Danish survey of 634,272 women with 1221,546 pregnancy outcomes, yielding an estimated overall fetal loss of 13.5% [8]. Of relevance is the observation that after age 40 the proportion of spontaneous abortions rises precipitously, exceeding 50%: the risk of spontaneous abortion has been estimated at 8.9% in women aged 20–24 years, in stark contrast to a whopping 74.7% in those aged 45 years or more [8]. Increasing paternal age seems also associated with a small increase in spontaneous abortion, with a pooled risk for age category ≥45 years of 1.74 (95% CI = 1.26–2.41) compared to the age group 30–35 [9].

The existence of a ‘black box’ was first documented in the mid and late 1950s when precious information on early embryonic losses became available with the publication by Hertig and his group [10,11,12] of the results of a careful investigation of 34 human embryos aged between 1 and 17 days (8 were considered to be in the preimplantation stage), collected over a period of 17 years. They found that 4 of the 8 preimplantation embryos presented with such severe anomalies that gestation could not have proceeded to term. In addition, 6 (23%) of the embryos in the second or third week also had anomalies such as to be incompatible with normal development. In 1973, Hertig and Rock [12] clearly identified the period during which a major loss of early embryos occurs as the first 2 weeks after fertilization (i.e., prior to the first missed period) and described a phenomenon they coined ‘*disintegration of an ovum once fertilized*’. This implies that data on spontaneous abortion provide only a very partial view of the fate of fertilized human oocytes, with substantive additional information coming from a variety of disciplines.

The main scope of this review is to highlight the clear distinction in pregnancy wastage between the first 2 weeks after conception and the subsequent 9 months of gestation; we then argue that, in light of this distinction, a new nomenclature specifically separating the two periods should be adopted.

## 2. Methodology

The availability of several reviews of the subject provided the basic set of references for the present work. In addition, a PubMed search using the words “early pregnancy wastage” yielded 241 references. More were added using references listed under “similar articles” after each relevant entry. A series of specific searches were also carried out with a variety of expressions: “early pregnancy factor” (215 entries); “pre-implantation factor” (931 entries, increasing to 2136 if using the acronym “PIF” and to 4714 when using “preimplantation factor); “embryo-derived Platelet-activating Factor” (48 entries); and “immunological factors in early pregnancy wastage” (33 entries).

The wording “endometrial factors in implantation” yielded 3193 titles; these decreased to 2436 when adding the word “human”. Therefore, the search was undertaken manually, with the ensuing, albeit remote, possibility of omission.

(All entries as accessed on 1 August 2023).

## 3. Evidence of Early Embryonic Loss

In 1975, Roberts and Lowe were the first to attempt a statistical estimation of pregnancy wastage, placing emphasis on early losses. Their conclusions were somewhat shocking: they estimated that at least 75% of all conceptions do not become a viable fetus and proceed to term [13]. A few years later, Shepard and Fantel quoted an early embryonic/fetal loss of approximately one out of two pregnancies and attributed it to ‘*karyotype deviation*’ [14].

The availability of the so-called ‘Rosette Inhibition test’ to evidence the presence of a pregnancy-specific protein named *Early Pregnancy Factor* (EPF) [15], enabled Rolfe to monitor fertilization and early gestation in a group of 13 nulliparous women during 28 cycles. In 18 subjects the presence of EPF was detected in maternal serum within 48 h of the presumed fertilization, but EPF production continued for more than 14 days in only 4 cases and successful pregnancy was maintained in only 2. In the remaining 14 cases, EPF disappeared from the serum before the presumed time of onset of menstruation [16]. A similar investigation was conducted on 18 healthy women during 21 menstrual cycles. EPF was present in 14 cycles, with 6 showing only a transient activity over a 5–10-day period following ovulation [17]. Once again, these investigations showed a surprisingly high proportion of early embryonic loss.

In 1988, Wilcox et al. [18] utilized a highly specific immunoradiometric assay with a sensitivity to detect urinary human chorionic gonadotrophin (hCG) of 0.01 ng/mL, to evaluate the risk of early pregnancy loss in 221 healthy women attempting to conceive over a total of 707 menstrual cycles. They identified 198 pregnancies with an increase in the hCG level near the expected time of implantation and observed that 22% of them ended before pregnancy could be detected clinically. Of relevance, most of the women with unrecognized early pregnancy losses had normal fertility, and, indeed, 95% of them became clinically pregnant within two years.

Demographic and epidemiological evidence also calls for the presence of a major early wastage. Indeed, human fecundity, i.e., the capacity to bear live children, defined as ‘*the probability to produce a vital term newborn per menstrual cycle during which there was normal sexual activity*’ [19], rarely exceeds 35–40% [4]. In fact, Woods estimated that apparent fecundability varies from 0.14 to 0.31 (0.17–0.38 when adjusted for fetal loss) [20].

A recent investigation of Japanese couples trying to conceive their first child utilized the “time to conception” (TTP) as either “natural” or “total”. Using TTP-total and TTP-natural, the sterile proportion of the whole sample was, respectively, 2% and 14%, and the interquartile range of fecundability (IQRF) was, respectively, 0.10 (CI: 0.04, 0.19) and 0.11 (CI: 0.05, 0.19). The IQRF was 0.18 (CI: 0.10, 0.29) for women aged 24 years or younger and 0.05 (CI: 0.02, 0.13) for 35–39 years old when TTP-all was used; the TTP-natural was 0.18 (CI: 0.10, 0.29) for women aged 24 years or younger and 0.06 (CI: 0.00, 0.15) for 35–39 years old. This investigation concluded that fecundability is overall lower at higher ages, while interquartile ranges are overlapping [21].

A summary of information on fecundability in the human available up to 2010 is presented in Table 1.

Reasons for the reported wide variations in evaluating human fecundity have been critically examined by Smarr et al. [30] who pointed out that there is no ‘population (bio)marker’ that can be used and, as such, fecundity can only be assessed indirectly utilizing a variety of individual- or couple-based endpoints, defaulting both evaluation and monitoring to rely on rates of births (fertility) or adverse outcomes.

In terms of possible causes for the relatively low human fecundity, there is once again increasing evidence that it is due to pre-clinical pregnancy loss, suggestive of spontaneous failure of implantation. Therefore, the mechanisms underlying this phenomenon need to be critically analyzed.

In a 1996 clinical study, Zinaman et al. found a maximal fertility rate of approximately 30% per cycle in the first two cycles, with a pregnancy wastage of 31%. Importantly, 40% of these losses occurred so early after fertilization that the presence of gestation could be identified only through the measurement of urinary hCG [29]. Another clinical investigation followed a cohort of 217 women attempting to become pregnant, observing early loss rates ranging from a low estimate of 11.0% to a high one of 26.9% [31]. More recently, Leridon provided an average figure of 20–25% (at ages 20–30) and pointed out that, although human fetal mortality is high, amounting to 12–15% of confirmed pregnancies, “*an even higher proportion of ‘products of conception’ do not develop normally and are evacuated within a few weeks, before the woman becomes aware of her pregnancy*”. He cites a figure of around 50% of all conceptions and states that in their great majority these failures are due to severe genetic abnormalities, concluding that “*human reproduction has a high error rate, but most of these errors are corrected by eliminating the products of conception*” [32].

A number of investigations have attempted an evaluation of pre-clinical embryonic losses in in vitro fertilization (IVF) cycles.

At the turn of the millennium, Simon et al. [33] recruited 145 subjects undergoing IVF and 92 undergoing oocyte donations. In subjects undergoing IVF, positive implantation was documented in 60.7% of embryo transfer (ET) cycles, but only 20.7% resulted in viable pregnancies, whereas the remaining miscarried; this occurred at an early pre-clinical stage in 72.4% of the failed ET cycles. In ovum donation cycles, positive implantation was recorded in 69.6% of ET cycles. Of these, 37.0% miscarried; among them, pre-clinical losses accounted for 70.6% of the total.

Boomsma et al. [34] utilized rates of rising urinary hCG (indicating initiation of implantation) that did not lead to a subsequent positive pregnancy test. They found that in over 50% of women undergoing ET, a rise in hCG could be documented, indicating an implanting embryo. However, in approximately one-third of these implanting embryos, a pre-clinical pregnancy loss occurred.

Calculations of early embryonic losses have been scrutinized by Jarvis [35], who expressed the opinion that: (i) the hypothesis by Roberts and Lowe [13] has no practical quantitative value; (ii) life-table analyses cannot evaluate losses at very early stages of development; (iii) measurement of hCG can only reveal losses occurring from the second week of gestation; and (iv) calculations by Hertig and his group [10,11,12] are highly imprecise and cast doubt on the validity of their analysis.

In support of his views, Jarvis quotes the large variations of losses published in a number of reputable scientific publications, before and during implantation (30–70% [36], >50% [37], and 75% [38]). He then mentions his re-analysis of the results of hCG investigations and concludes that approximately 40–60% of embryos may be lost between fertilization and birth. His critique of the work of Hertig’s group is based on the fact that all estimates of early embryo mortality are subject to commensurate inaccuracy in the absence of reliable fertilization probabilities which, as stressed by Short [39] are “*surprisingly difficult to estimate*”.

Jarvis’ critical reanalysis has merit, but—according to his own revision of published data—early pregnancy wastage would still reach at least 50%.

Finally, in 2020, Wilcox et al. [40], after combining data from epidemiologic, demographic, laboratory, and in vitro fertilization investigations, constructed an empirical framework aimed at producing plausible estimates of fecundability, sterility, transient anovulation, patterns of intercourse, and the proportion of ova fertilized in the presence of sperm. After combining all this information, they generated an estimation of preimplantation loss, to be considered an average for fertile couples, concluding that “*under a plausible range of assumptions… 40 to 50% of fertilized ova fail to implant*” even in normally fertile couples.

## 4. Mechanisms Involved in Early Embryo Selection

The first week of development is characterized by daily changes that can be summarized as follows:

Day 1: Fertilization takes place with the fusion of the sperm and egg to form the *zygote*.

Day 2: The division of the zygote begins, and two cells are formed.

Day 3: A solid spheric ‘ball’ of cells, the *morula*, is generated by the further division of the zygote.

Day 4: The morula continues to divide, while in its center a small cavity begins to form as the morula transforms into the *blastocyst*.

Day 5: The blastocyst begins to implant in the uterus.

Day 6: The implantation process continues, and the blastocyst begins to differentiate with the formation of an outer layer: the trophoblast.

Day 7: Implantation is complete, with the blastocyst fully embedded in the decidua; at this stage, the process of placental formation begins and is characterized by the formation of the two layers: cytotrophoblast and syncytiotrophoblast.

In 2017, Makrigiannakis et al. [41] summarized the steps involved in the process of implantation as follows:The blastocyst moves towards the uterine cavity and contemporarily the decidualized endometrium evolves to a receptive phenotype, resulting in a biochemical cross-talk with the embryo.The pre-implantation embryo begins to secrete factors capable of modulating the implantation site while, in turn, the decidua secretes cytokines and growth factors modulating embryonic differentiation and development.In the presence of a proper biochemical environment, the embryo and the decidua jointly promote trophoblast invasion. This process can be altered in a number of ways resulting in early embryonic loss.

A schematic view of the major phases of the implantation process is presented in Figure 1.

Over the last decades, information has been obtained concerning the mechanisms leading to this very early embryonic demise. These have been clearly summarized by Macklon and Brosens [43], who proposed that maternal systems are in place to prevent the implantation of poorly viable embryos. To achieve this, the decidualization of the endometrium acts as a biosensor through which signals from the early embryos are converted into a ‘go’ or ‘no-go’ endometrial response and therefore successful implantation.

Brosens et al. [44] have further elaborated the path that led to new insights into the processes that govern maternal selection of human embryos in early gestation, pointing out the major challenge for the maternal organism (and specifically for the decidualized endometrium) to eliminate defective embryos without at the same time preventing implantation of normal blastocysts. Two distinct mechanisms seem to be at work: the first implies that a proportion of embryos simply fail to implant; the second is that they are rejected soon after the beginning of the process of implantation in the endometrial luminal epithelium [45].

### 4.1. Factors Secreted by the Pre-Implantation Embryo

Fundamental new knowledge has been gained thanks to assisted reproduction techniques. An early investigation by Shutt and Lopata [46] of embryos cultured in vitro over a period of 3–4 days, found that the corona cells surrounding the fertilized ovum could secrete daily a mean amount of 50 ng of progesterone and approximately 100 pg each of estradiol, prostaglandin-E_2_ (PGE2), and PGF2α, respectively. This experiment provided the first evidence of how the early embryo is supplied with vital substances for its survival before the maternal organism intervenes.

After the blastocyst reaches the uterine cavity, several substances of embryonic origin become involved in the intense cross-talk between the early embryo and endometrium, initiating before and conditioning the process of implantation and of the placenta’s formation.

#### 4.1.1. The Early Pregnancy Factor

In the 1970s, Morton et al. [47], starting from the observation that human lymphocytes showed a depression in their activity when incubated in serum from pregnant women, identified the above-mentioned EPF. Biochemically, EPF is a homolog of chaperonin 10 and belongs to the heat shock family of proteins with immunosuppressive and growth factor properties [48] and seems involved in the suppression of the maternal response, thereby allowing the continued viability of the early embryo. Nahhas and Barnea [49] believe that EPF represents a link between fertilization and immunomodulation and that during the pre-implantation period, EPF is of maternal origin, whereas after nidation it becomes of embryonic origin.

#### 4.1.2. The Pre-Implantation Factor

A second early-secreted substance, coined *Pre-Implantation Factor* (PIF), was identified by Barnea et al. [50] in 1994. Starting from the observation that in men and non-pregnant women, the proportion of lymphocytes bound by platelets is significantly different from that of pregnant women (*p* < 0.0001), they developed an assay to measure the presence of PIF. This was detected in subjects who had successfully undergone IVF-ET, followed by a normal pregnancy by 4 days after transfer. PIF is a linear peptide molecule consisting of 15 amino acids, which exhibits a self-protective and antitoxic action and is present in maternal blood as early as 7 days after conception and absent in the presence of non-viable embryos [51].

Yang et al. [52] identified 21 proteins capable of interacting with PIF. Of particular interest is myosin heavy chain 10 (MYH10), since silencing its expression in human endometrial adenocarcinoma (HEC-1-B) cell culture, significantly attenuates cell migration and invasion capacities. It seems therefore that MYH10-mediated cell migration and invasion act in conjunction with PIF in promoting trophoblast invasion. PIF has different biological functions in mammalian species and plays a major role in the embryo’s neural system development and neuroprotection; updated information obtained from evidence-based studies as of 2020 is available [53].

Finally, it has been recently suggested that PIF accentuates the decidualization process and the production of endometrial factors that limit trophoblast invasion. Therefore, by controlling both trophoblast and endometrial cells, PIF seems to play a major role in the process of human embryo implantation [54].

#### 4.1.3. The Embryo-Derived Platelet-Activating Factor

A third early factor, the *embryo-derived platelet-activating factor* (PAF) (1-o-alkyl-2-acetyl-sn-glycero-3-phosphocholine) was identified by O’Neill et al. [55]. Subsequently, Roudebush et al. [56] proved a correlation between PAF levels in human embryo culture media and pregnancy outcome: as PAF levels increase, so does the corresponding pregnancy rate.

The activity of PAF is regulated at the level of its synthesis and degradation as well as by the expression of a specific cell surface receptor (PAFr); PAF is eventually degraded by the enzyme PAF-acetyl-hydrolase (PAF-AH). It has been suggested that PAF may constitute the embryonic signal controlling embryo transport to the uterus. Velasquez et al. [57] have documented the co-localization of both PAFr and PAF-AH in the epithelium and stromal cells of the fallopian tubes, granting credibility to this theory.

In 2005, O’Neill summarized knowledge of the factor [58], noting that PAF is produced and released by the embryos of all mammalian species studied to date and is considered the best-described embryotrophin, with a role in the ligand-mediated trophic support of the early embryo.

Sato et al. [59] histologically investigated the aggregates of trophoblasts invading spiral arteries and observed deposition of maternal platelets in them. Furthermore, there is evidence that these platelets are activated and, in an in vitro system, they are capable of increasing the ability of trophoblast to invade spiral arteries.

Recently, PAF has been identified as a fetus-derived mediator repressing placental progesterone receptor A (PR-A) in the human placenta leading to the activation of pro-labor signaling. It seems therefore that PAF represents a novel fetal-derived candidate for initiation of labor [60].

#### 4.1.4. The Human Chorionic Gonadotrophin

In spite of the progress made in the biology of these pre-implantation factors, the best known and the most important among them still remains hCG, indispensable for maintaining the corpus luteum. Fishel et al. [61] were the first to detect its presence in the medium surrounding two embryos cultured for more than 7 days after IVF. This places hCG among the pre-implantation factors. In fact, its mRNA is transcribed as early as the 8-cell stage [62].

### 4.2. Factors Produced by the Endometrium

Successful implantation requires the exquisitely coordinated migration and invasion of trophoblast cells from the outer capsule of the blastocyst into the endometrium [42]. Over and above the described substances produced by the early embryo, the process is also guided by additional signals from cells in the decidualized endometrium. There are numerous recent descriptions of the mechanisms involved in successful implantation, e.g., [3,32,34,36,39,42,43,56,57,63]. For this reason, a detailed analysis of the implantation process is beyond the scope of this review.

Briefly, however, for the implantation process to be successfully achieved, a well-orchestrated interplay of various cell types is necessary. These consist of epithelial, stromal, and immune cells, and trophoblasts, producing a number of cellular molecules, such as cytokines, chemokines, growth factors, and adhesion molecules [42,63]. In addition, endometrial stroma cells have been shown to promote trophoblast invasion through the generation of an inflammatory environment modulated by TNF-α (tumor necrosis factor α) [64]. Furthermore, through the production of IL-17 (interleukine-17), stromal cells promote trophoblast migration [65]. Finally, adequate glycolysis also appears to be necessary to provide all the energy and macromolecules needed for implantation and early pregnancy [66].

### 4.3. The Role of the Immune System in Early Pregnancy Wastage

Some twenty years ago, Clark, starting from the fact that embryos bear paternal and embryonic antigens foreign to the maternal immune system, asked the question: could some otherwise normal embryos be ‘rejected’? [67]. In a subsequent review [68], he searched and critically analyzed the evidence in animals and humans, finding that various treatments may improve the live birth rate. Unfortunately, not enough evidence exists to indicate when in gestation such mechanisms may be active.

Early investigations by Haddad et al. [69,70] provided some experimental evidence in a murine model of the presence of activated macrophages at implantation sites before overt embryo damage occurs. In terms of mechanism, they showed that increased nitric oxide production by decidual macrophages is involved in early murine embryo loss.

Vento-Tormo et al. [71] have determined the cellular composition of human decidua, documenting the existence of subsets of perivascular and stromal cells located in distinct decidual layers. These include immune cells, principally uNK cells (~70%), macrophages (~20%), and T cells (~10%). In a mouse model, the influx and expansion of these immune cells are controlled by decidual cells, since effector T cells cannot accumulate within the decidua, thanks to the epigenetic silencing of key T cell-attracting inflammatory chemokine genes in decidual stromal cells [72]. According to Brosens et al. [44], cooperation between different innate immune cell populations is essential for optimal decidualization. Therefore, an adverse impact of impaired decidualization on local immune populations and vice versa, hampering the distinction between cause and effect.

A recent review of endometrial immune dysfunction in recurrent pregnancy loss [73] found experimental and clinical evidence suggesting that derangement of the endometrial immune environment can be involved in recurrent implantation failure. Indeed, both changes and modulation of the activity of the immune cells at the level of the endometrium occur early in gestation and the maternal immune system involvement is extended to the endometrial tissue breakdown, vascular remodeling, and placentation. What is not clear is whether the involvement of the immune system occurs at the very early pre-implantation stage, although there are data in support of the hypothesis that the process of nidation evokes an initial, early, inflammatory reaction which is promptly followed by the establishment of an anti-inflammatory decidual environment, allowing the survival of the conceptus and the progression of the pregnancy.

In addition, regulatory T cells (Tregs) also seem to play a vital role in implantation and sustaining pregnancy [74]. The fact that a healthy woman can successfully carry her genetically disparate fetus to term without immune rejection strongly indicates that immune cells are actively involved in pregnancy. Tregs are thymus-derived and can be recruited to non-lymphoid tissues to curtail inflammation and maintain immunological self-tolerance and homeostasis [75]. Tregs may also differentiate and proliferate from CD4+ naive T cells when stimulated with immunosuppressive cytokines such as TGF-β and IL-10 [76]. The potent immunosuppressive activity of Tregs comes from their ability to influence the polarization, expansion, and effector function of T effector cells [77]. Tregs can inhibit effector immunity, restrict inflammation, and support maternal vascular adaptations, and, as such, facilitate trophoblast invasion and placental access to the maternal blood supply. Ample data have shown that insufficient Treg numbers or inadequate functional competence are involved in idiopathic infertility and recurrent miscarriage, as well as later-onset pregnancy complications, ranging from preeclampsia and fetal growth restriction [78].

## 5. Factors Predictive of Embryonic Loss

### The Embryo–Maternal Dialogue

In reviewing the situation with PIF and other early factors, Barnea [79] pointed out that the embryo–maternal dialogue starts shortly after fertilization and is exerted through both local and systemic signaling. First comes the PIF, secreted in humans already at the 2–4-cell stage, that initiates the modulation of cellular immunity. Then comes a class of novel proteins/peptides coined *developmental proteins* (DPs), present in the embryo before a mature immune system has developed. They promote normal proliferation while controlling any abnormal one and seemingly acting through specific receptors. When the development of an embryo becomes incompatible with life, DPs may lead to growth arrest, a decline of pre-implantation factors, reactivationof the immune system, and, ultimately, pregnancy rejection. Drawing attention to the fact that, despite the use of adjuvant therapies, the cumulative rates of live births following IVF-ET remains at ~40%, Yang et al. [52] stressed that the low pregnancy rates, even in the presence of high fertility rates, are due to implantation failure. As reported above, they tried to construct a profile of the DPs reacting with PIF in an attempt to clarify the molecular mechanisms by which PIF promotes trophoblast invasion (see Section 4.1.2).

Brosens et al. [44] have identified multiple decidual checkpoints conditioning the implantation of the blastocyst. The first is appropriate timing since a delayed rise in hCG levels beyond the putative implantation window is strongly associated with the interruption of gestation in the first two weeks of pregnancy. The second is the action of migratory decidual cells encapsulating the implanting embryo, serving as biosensors, and acting in both negative and positive selection. In the presence of low-quality embryos, these cells engage in a stress response inhibiting the secretion of implantation factors and hindering embryo encapsulation, while normal embryos secrete factors enhancing the expression of maternal implantation and metabolic genes, thus actively promoting the completion of the implantation process. The third is represented by a sufficient secretion of hCG to rescue ovarian progesterone production until around 8 to 10 gestation weeks when the placenta takes over progesterone production [80].

Brosens et al. [44] concluded that at the time of implantation, both positive and negative selection mechanisms are active in a process that is intrinsically dynamic and adaptable (Figure 2).

Information is also available on the mechanisms through which the early embryo identifies its own defects, such as, first and foremost, apoptosis which begins to appear at the blastocyst level [81,82]. The starting point has been the observation that many embryos grown in vitro contain unequal-sized blastomeres and multiple cellular fragments and, when fragmentation becomes excessive, their developmental potential both in vitro and in vivo is severely limited [82]. Hardy [83] reported that embryos cultured in vitro, when evaluated after their developmental arrest, often show features characteristic of apoptosis, whereas embryos that seem to be developing normally show no such features before compaction.

A recent review indicates that the most studied mechanisms of embryo fragmentation are apoptotic cell death, membrane compartmentalization of altered DNA, cytoskeletal disorders, and vesicle formation. These phenomena may result in the extrusion of entire blastomeres, the release of apoptotic bodies and other vesicles, and the formation of micronuclei [82]. Indeed, back in 1996, Jurisicova et al. [81] evaluated arrested, fragmented human embryos and were able to detect extensive condensation and degradation of chromatin, which is suggestive of apoptosis. Of importance, no such abnormalities were observed in embryos with regular-sized blastomeres and absence of fragmentation; these findings provided evidence for the existence of the mechanism coined *programmed cell death*. In human embryos, such a mechanism seems to be triggered at a stage prior to blastocyst formation, leading to preimplantation embryo death.

According to Leidenfrost et al. [84], in the bovine model, errors and even failure of the first few cleavage divisions frequently cause immediate embryo death, which constitutes the main source of developmental heterogeneity. This seems to be due to a systemically and developmentally controlled elimination of cells, while the nature and mechanisms of the inner cell mass cell death are unclear.

An early investigation by Liu et al. [85] failed to find a significant difference in the expression frequency of all studied genes between viable and nonviable or arrested embryos, but with one exception: *BCL-2* was only detected in viable embryos. Shortly after, Jurisicova et al. [86] reported that human oocytes and preimplantation embryos possess abundant levels of transcripts encoding cell death suppressors and apoptosis inducer genes, *Bax* and *Caspase-2*. They also conducted an analysis of gene expression in single human embryos exhibiting various degrees of fragmentation at the 2-, 4-, and 8-cell stages and reported that at the 4-cell stage, embryos displaying 30 ± 50% fragmentation showed a significant increase in the *Harakiri* gene and *Caspase-3*, suggesting that cellular fragmentation could be regulated by certain components of a genetic program of cell death. They stressed that cell fate (i.e., survival/differentiation or death) is determined by the outcome of specific intracellular interactions between pro- and anti-apoptotic proteins, many of which are expressed during oocyte and preimplantation embryo development [87].

The same group [88] also examined changes in mitochondrial membrane potential over the preimplantation stages of mouse and human embryos. They found that mouse zygotes and early embryos contain highly-polarized mitochondria and observed a transient increase in the ratio of high to low ΔΨm (delta psi m, a measure of the mitochondrial membrane potential) with increasing cleavage. In human 8-cell embryos, there was an increased ratio of high- to low-polarized mitochondria, as well as increasing degrees of embryo fragmentation, leading to the conclusion that an aberrant shift in ΔΨm is either associated with or can contribute to a decreased developmental potential. Komatsu et al. [89] measured the ΔΨm of the in vivo-fertilized 1- and 2-cell stage, and of IVF embryos and found that the ΔΨm of in vivo-fertilized embryos was highly upregulated, whereas in a number of IVF embryos remained unchanged. In addition, the development of low-ΔΨm 2-cell stage IVF embryos tended to be arrested after the 2-cell stage. Zhao et al. [90] tried to profile the role of the mitofusin2 gene *Mfn2* (a key player in many mitochondrial activities, such as fusion, trafficking, turnover, and contacts with other organelles [91]) in mouse embryos and determine the underlying mechanism of *Mfn2* function in embryo development. They concluded that low in vitro expression of *Mfn2* causes mitochondrial dysfunction and attenuates blastocyst formation rate. These findings indicate that *Mfn2* could affect preimplantation embryo development through mitochondrial function and cellular apoptosis [90].

Recently, a study by Haouzi et al. [92] of genes involved in the regulation of the apoptotic and survival pathways of mouse and human embryos found that components of the major apoptotic and survival signaling pathways were expressed during early human and mouse embryonic development in a species-specific manner.

According to Brosens et al. [44], genome-wide screening of blastomeres in IVF cycles seems to harbor cells with complex large-scale structural chromosomal imbalances, mostly by mitotic non-disjunction. In addition, a vast array of chromosomal errors has been detected in human embryos throughout all stages of pre-implantation development [44].

In an attempt to identify visual markers of early pregnancy loss in women undergoing IVF, Amitai et al. assembled an expansive set of 314 morphological, morphokinetic, and dynamic features derived from measurable static and dynamic properties of preimplantation embryo development [93]. They identified a subset of six non-redundant morphodynamical features possessing high predictive capacity. Among them, of particular interest were features that account for the distribution of the nucleolus precursor bodies within the small pronucleus and pronuclei dynamics. Using these features, they developed a “decision-support tool” for prioritizing embryos for transfer based on their predicted implantation potential.

The challenge now is to find a way to apply the identification of these markers to natural pregnancies.

## 6. The Need for New Terminology

As summarized here, human preimplantation embryos exhibit in vitro high levels of apoptotic and high rates of developmental arrest during the first week [82]. In fact, in vitro and in vivo, errors and failures of the first and the next three cleavage divisions frequently cause immediate embryo death or lead to aberrant subsequent development. At the same time, the ability of human embryos to eliminate/expel abnormal blastomeres as cell debris/fragments and carry out—whenever possible—self-correction or quality control has been also documented [94]. Finally, Monsivais et al. [95] have shown that for the correct regulation of uterine receptivity, a convergence is necessary of bone morphogenetic proteins (BMPs, members of the TGF-β family that regulate the post-implantation and mid-gestation stages of pregnancy) and steroid hormone signaling pathways.

For a number of years, in assisted reproduction, the establishment of a viable pregnancy has been defined by a rise in circulating hCG. The use of this marker shows that in IVF the successful implantation of an embryo represents the major milestone in determining the success of gestation since it has been shown that only 50% of transferred embryos implant and that half of these embryos are subsequently lost [34].

The documented occurrence of massive early/preimplantation embryo loss provides the scientific basis to necessitate a clear distinction between the first 1 or 2 weeks and the subsequent 9 months of gestation. In the first phase, a physiological loss of fertilized oocytes/early embryos, calculated at around 50%, occurs, whereas during the second phase, the pathologic wastage of embryos/fetuses has been estimated to be around 15%, although increasing with maternal age. Given this stark contrast, it seems appropriate to distinguish these two periods also using a different nomenclature.

This is why we fully concur with the idea of, on the one hand, employing the word **gestation** to identify the period from fertilization (whether in vitro or in utero) to birth, and, on the other, to utilize the word **pregnancy** when referring to the period after implantation is completed [1]. The idea is certainly not new in view of the above-mentioned definition of pregnancy in the IVF clinic as the rise of hCG upon embryo implantation [96]. Extending this concept to all pregnancies would create a simple, clear nomenclature.

This seems the position taken by several organizations, which—in practice—affirm that there is no pregnancy before implantation: among them is the World Health Organization [97], which, referring to “Medicines for Reproductive Health and Perinatal Care” (Item 22), under ‘*Oral Hormonal Contraceptives*’ (22.1.1), lists hormonal emergency contraceptives levonorgestrel and ulipristal; and the other is the American College of Obstetricians and Gynecologists [98].

## 7. Conclusions

The continuing progress in our understanding of the complexity of interactions between the maternal organism and the early embryo is changing our overall outlook on the initial steps in establishing a pregnancy through placental formation. The first two weeks after fertilization must today be viewed as the critical period during which a major embryo selection process takes place in which a proportion that may surpass 50% of them is physiologically eliminated because they are unfit to progress toward birth.

The new knowledge, along with a refined nomenclature to distinguish gestation and pregnancy, should help us in improving the effectiveness of the various assisted reproduction technologies, as well as providing the scientific basis for a clear distinction between the first two weeks following fertilization and the rest of gestation.

## Figures and Tables

**Figure 1 ijms-24-15420-f001:**
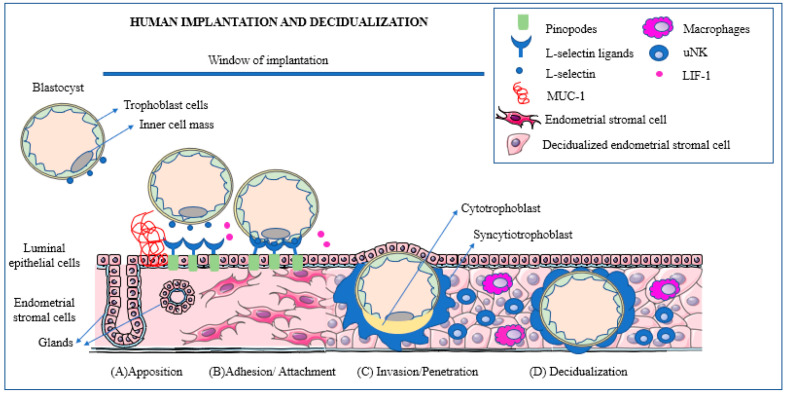
Schematic representation of the main phases of human implantation: apposition, adhesion/attachment, invasion/penetration, and decidualization. (**A**) (Apposition): the blastocyst expresses L-selectin that reacts with its ligands. The presence of Mucin-1 repels the blastocyst and prevents it from attaching outside of the window of receptivity. (**B**) (Adhesion): the blastocyst promotes cleavage of Mucin-1 at the implantation site to ensure successful attachment. (**C**) (Invasion): blastocyst trophoblast cells penetrate the endometrial epithelium and reach the stroma. As soon as implantation is initiated and the embryo reaches the stromal cells surrounding the embryo, these transform into decidualized cells. (**D**) (Decidualization): during the decidualization process, immune cells, such as macrophages and uterine natural killer (uNK) cells, play an important role promoting an environment conducive to successful implantation. Reprinted with permission from: Ochoa-Bernal and Fazleabas (2020) [42].

**Figure 2 ijms-24-15420-f002:**
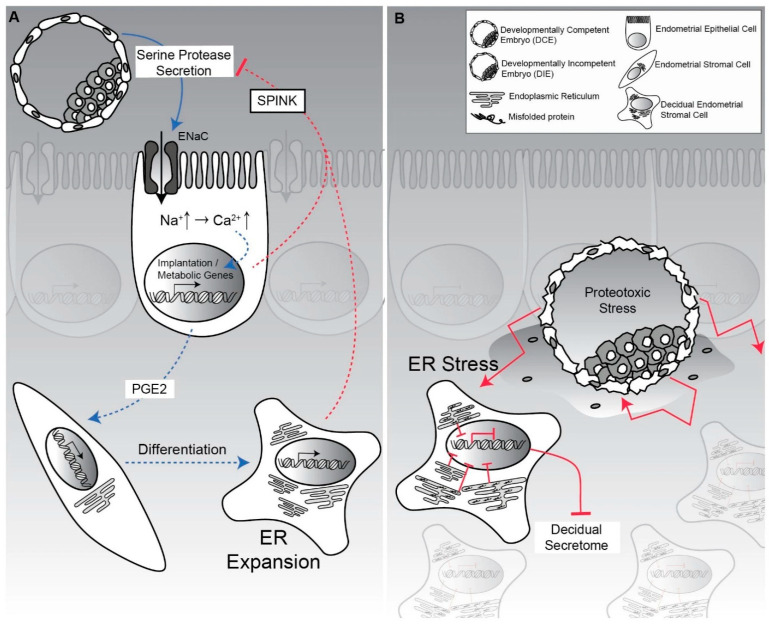
Positive and negative mechanisms contribute actively to the selection of human embryos at implantation. (**A**) Developmentally competent human embryos secrete serine proteases that activate epithelial Na1 channel (ENaC) that induce genes involved in implantation and post-implantation embryo development. (**B**) Developmentally compromised embryos By contrast, induce excessive protease activity, leading to accumulation of misfolded proteins and ER stress, compromising decidual cell functions and triggering early maternal rejection. From: Brosens JJ et al. (2014) [44].

**Table 1 ijms-24-15420-t001:** Human fecundity over the last four centuries.

Population Studied	Publication	Fecundity Index
Canada (Québec), 18th century	Henrypin (1954) [22]	0.31
China 20th century	Wang et al (2003) [23]	0.40
France, 17th and 18th centuries	Charbonneaux (1970) [24]	0.21
Great Britain, 20th century	Vessey et al. (1976) [25]	0.21
Mexico, 20th century	Balakrishnan (1979) [26]	0.21
Peru, 20th century	Balakrishnan (1979) [26]	0.17
The Netherlands, 20th century *	van Noord Zaadstra et al. (1991) [27]	0.54
USA (Hutterite sect) 20th century	Sheps (1965) [28]	0.28
USA, 20th century	Zinamen et al. (1996) [29]	0.30

* After 2 months at age 31. Modified and reprinted with permission from Benagiano et al., 2010 [4].

## Data Availability

Not applicable.

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
