# Peer review of "Events Leading to the Establishment of Pregnancy and Placental Formation: The Need to Fine-Tune the Nomenclature on Pregnancy and Gestation"

_ijms, 2023, doi:10.3390/ijms242015420_

Round 1

Reviewer 1 Report

The authors have done a great job work for this review. I would like to point out, that this review provides a systematic collection of the current literature.

The novelty of this study is that the authors precisely define the first two weeks after fertilization and suggest distinguishing strictly between the period from fertilization to birth and the period after implantation and pregnancy.

Minor comments:

Comment 1:

Line 80-83: I would like to ask the authors, why they mention that intercourse during the implantation window might result less likely in a positive pregnancy without any explanation or any other context of this sentence.

Comment 2:

Line 328: the authors have written a whole paragraph about PAFs without mentioning the relevance for platelets (as the name implies) and the relevance of platelets for the implantation. Maybe they can include some sentences on that topic.

Reviewer 2 Report

Study is about a new terminology proposal for early pregnancy periods. Mechanisms from the firsts two weeks are clearly described and therefore your manuscript brings a new point of view regarding first trimester abortion, a well known pathology.

Figure 1 is a low resolution image and should be changed to a more clear one.

A more clear take home message is conclusion section should be described apart from a nomenclature regarding clear distinction between the first two weeks following fertilization and the rest of gestation.

Reviewer 3 Report

Thanks to the authors of this very interesting and exciting manuscript 'Events Leading to the Establishment of Pregnancy and Placental Formation: The Need to Fine-Tuning the Nomenclature on Pregnancy and Gestation'.

Yet, I would like to give some comments:

1. factors that are contributing to successful pregnancy could be reported in a separate table; as an alternative, you could try to add them into fig 1.

2. include cell origin by listing secretion of different products and which cells are affected in which way;

3. fig. 1 is hazy. try to sharpen it!

4. in the abstract ll 42-48: to my knowledge, the placenta is already implanted within day 5/6 pc, and the primary placental villi are already developed on day 13/15, secondary villi developed day 15/16, tertiary villi developed day 17/18. meaning that villous morphogenesis takes place  between day 13-21. ref: M. Vogel, G. Turowski. DeGruyter 2019: 'Clinical Pathology of the Placenta', 

This means, that placenta formation takes place from implantation on day 5/6, meanig week 1.

5. timeline of embryonic development (blastcyst) should be included in your text and argumentation already in the abstract;

6. ll 49-113: much, mentioned in literature belongs to the discussion, should not be part of the introduction; 

7. l 245: correct: 'selection';

8: suggestion: make '4.3. The Embryo-Maternal Dialogue' to nr. 5;

9: suggestion: make '4.4. The Role of the Immune System in Early Pregnancy Wastage' to nr. 6;

10.  'The Need for New Terminology'....could be part of the discussion;
